# Validation of the Conventional Trauma and Injury Severity Score and a Newly Developed Survival Predictive Model in Pediatric Patients with Blunt Trauma: A Nationwide Observation Study

**DOI:** 10.3390/children10091542

**Published:** 2023-09-12

**Authors:** Chiaki Toida, Takashi Muguruma, Masayasu Gakumazawa, Mafumi Shinohara, Takeru Abe, Ichiro Takeuchi

**Affiliations:** 1Department of Emergency Medicine, Teikyo University School of Medicine, Tokyo 173-8606, Japan; 2Department of Emergency Medicine, Yokohama City University Graduate School of Medicine, Yokohama 232-0024, Japan; mgrmtks@gmail.com (T.M.); msys.gkmzw@gmail.com (M.G.); shinoharamafumi@yahoo.co.jp (M.S.); abet@yokohama-cu.ac.jp (T.A.); takeqq@yokohama-cu.ac.jp (I.T.)

**Keywords:** trauma scoring system, Trauma and Injury Severity Score, survival prediction model, children, pediatric patients with blunt trauma, Japan Trauma Data Bank

## Abstract

To date, there is no clinically useful prediction model that is suitable for Japanese pediatric trauma patients. Herein, this study aimed to developed a model for predicting the survival of Japanese pediatric patients with blunt trauma and compare its validity with that of the conventional TRISS model. Patients registered in the Japan Trauma Data Bank were grouped into a derivation cohort (2009–2013) and validation cohort (2014–2018). Logistic regression analysis was performed using the derivation dataset to establish prediction models using age, injury severity, and physiology. The validity of the modified model was evaluated by the area under the receiver operating characteristic curve (AUC). Among 11 predictor models, Model 1 and Model 11 had the best performance (AUC = 0.980). The AUC of all models was lower in patients with survival probability Ps < 0.5 than in patients with Ps ≥ 0.5. The AUC of all models was lower in neonates/infants than in other age categories. Model 11 also had the best performance (AUC = 0.762 and 0.909, respectively) in patients with Ps < 0.5 and neonates/infants. The predictive ability of the newly modified models was not superior to that of the current TRISS model. Our results may be useful to develop a highly accurate prediction model based on the new predictive variables and cutoff values associated with the survival mortality of injured Japanese pediatric patients who are younger and more severely injured by using a nationwide dataset with fewer missing data and added valuables, which can be used to evaluate the age-related physiological and anatomical severity of injured patients.

## 1. Introduction

Accurate survival prediction models for trauma patients facilitate quality control evaluation and trauma research as they enable valid comparisons of trauma populations with different degrees of baseline risk [1]. The Trauma and Injury Severity Score (TRISS) is the most commonly used method for calculating the survival probability in injured patients [2,3,4].

The coefficients of the current TRISS model were calculated based on the dataset from the Major Trauma Outcome Study coordinated from 1982–1987 by the American College of Surgeons Committee on Trauma [5]. In our previous study based on the current Japanese database, we showed that the TRISS methodology had a low performance and had underestimated the expected survivors in younger blunt trauma patients with a higher injury severity, because the in-hospital mortality of injured pediatric patients in Japan has decreased over the years [6,7]. Some studies have reported that the TRISS model is unsuitable for evaluating the survival outcome owing to its inaccuracy in terms of the area, era, and age of the study cohort [6,7,8,9,10]. Some studies have also suggested that local database-derived coefficients and variables may further enhance the accuracy of the TRISS model [11,12,13,14,15].

Moreover, in the TRISS model, the coded value of age is divided into two categories: <55 years (0) and ≥55 years (1). Furthermore, although patients with anatomical serious severity have a higher mortality risk, there is no universally acceptable definition for anatomical severity in injured patients. The most widely used definition of anatomical severity is the Injury Severity Score (ISS), which is based on an anatomical injury severity classification, an Abbreviated Injury Scale (AIS) [4,5]. Patients with an AIS ≥ 3 in at least two different injury regions had a worse outcome than those with an ISS ≥ 16 or ≥18 [16]. Therefore, in our proposed model, we have included age and number of injury region as new predictive variables.

To the best of our knowledge, there is no clinically useful prediction model for Japanese pediatric patients with trauma. We hypothesize that developing new coefficients and including predictive variables based on an analysis of Japanese nationwide database may improve the predictive accuracy for Japanese injured pediatric patients. This study aimed to developed a model for predicting the survival of pediatric patients with blunt trauma and compare its validity with that of the conventional TRISS model in Japan.

## 2. Materials and Methods

### 2.1. Study Design, Setting, and Population

In this retrospective, nationwide, observational study, we analyzed data obtained from the Japan Trauma Data Bank (JTDB) [12], which recorded the data of patients with trauma and/or burns and their prehospitalization and hospital-related information. Recorded data included demographic data, comorbidities, injury types, mechanism of injury, means of transportation, vital signs, AIS score, ISS, prehospital/in-hospital procedures, trauma diagnosis as indicated using the AIS, and clinical outcomes, as we previously described elsewhere [6]. Physicians were trained in AIS coding and registered an individual patient data [5,16]. As of March 2019, a total of 280 hospitals, including 92% of Japanese government-approved tertiary emergency medical centers, were participating [16]. The Japan Association for the Surgery of Trauma approves data access and updating of the registry, and the Japan Association for Acute Medicine performs the curation of all data [13].

Figure 1 summarizes the patient selection process. In this study, we utilized the JTDB dataset as described in the previous study [6]. Data were obtained for the period from 1 January 2009 to 31 December 2018. The inclusion criteria were as follows: patients with blunt trauma and age ≤ 18 years [6]. The exclusion criteria were as follows: age ≥ 19 years, the presence of burns or penetrating trauma, and out of hospital cardiac arrest, patients with AIS 6, or with missing outcome data and TRISS prediction. As we could not assess the physiological severity in patients with cardiac arrest on hospital arrival and the patients with AIS 6 are defined as the most serious anatomical condition who cannot survive, we excluded these patients. Among 26,329 blunt trauma patients younger than 18 years, 2480 patients (9.4%) had missing outcome, and 5421 patients (20.6%) were missing data for TRISS model. Accordingly, eligible patients were grouped into a derivation cohort who were registered in the JTDB between 1 January 2009 and 31 December 2013 and a validation cohort who were registered between 1 January 2014 and 31 December 2018 (Figure 1).

### 2.2. Data Collection

As previously described [8], data for the following variables from the JTDB was utilized: age (years), sex, Glasgow Coma Scale (GCS) score, systolic blood pressure (SBP), respiratory rate (RR), AIS, AIS of the injured region, Revised Trauma Score (RTS), ISS, Ps, and in-hospital mortality.

The TRISS score ranges from 0 (certain death) to 1 (certain survival), and the Ps is calculated as follows [2]:$$TRISS=Ps=\frac{1}{\left(1+e^{-b}\right)}$$
where
$$b=b0+\left(b1\times RTS\right)+\left(b2\times ISS\right)+\left(b3\times cAge\right)$$

RTS is calculated using the GCS score, SBP, and RR:$$RTS=\left(0.9368\times cGCS\right)+\left(0.7326\times cSBP\right)+\left(0.2908\times cRR\right)$$

The TRISS provides the probability of survival (Ps) based on the logistic regression model with predictor variables of the ISS, RTS, and categorized data of age year. The intercept and coefficients are determined by the AIS required for ISS calculation. Meanwhile, the RTS is calculated using the GCS score, SBP, and RR. The formula is a linear equation of their coded values as shown in Table 1. Furthermore, we selected the actual age and number of injury regions with AIS ≥ 3 as new predictive variables, as we hypothesized that adding new predictive variables may improve the predictive accuracy of the model.

### 2.3. Data Analysis

A logistic regression analysis was performed using a derivation dataset to establish new prediction models. First, we aimed to calculate the new coefficients of the TRISS model using logistic regression analysis based on the 10-year data obtained from the Japanese cohort (2009–2018). Second, we included the actual age year as a predictive variable for establishing the new prediction model. Finally, we included the number of injury regions with AIS ≥ 3 as a predictive variable in the proposed model.

The models developed in this study were described as follows
$$Logit\;\left(Ps\right)=\beta 0+(\beta 1\times Age)+(\beta 2\times Injury\;Severity)+(\beta 3\times Physiology)$$
where β0 denotes the intercept point; β1–3 denotes regression coefficients for the three predictors; age indicates age year; injury severity indicates ISS or number of injury regions with AIS ≥ 3; and physiology indicates RTS or physiological status (GCS score, SBP, or RR). Each model includes each of these injury severity and physiological status indicators as predictors. The coded values of age, GCS score, SBP, and RR are the same as those used in the TRISS and RTS [2]. The outcome was survival (=1) or death (=0) at hospital discharge. The Mann–Whitney U test and the Kruskal–Wallis test were used to analyze continuous variables. Pearson’s chi-square test was used for analyzing categorical variables. To predict survival, the accuracy of the developed models was evaluated in the validation cohort by calculating the receiver operating characteristic (ROC) curves that present the area under the ROC curve (AUC) and its 95% confidence interval [17]. The AUC represents the performance of each model for predicting the survivor of pediatric patients with blunt trauma. The highest is indicated by an AUC value close to 1, whereas a poor performance is indicated by an AUC near 0.

We have previously described our data analysis method [5]. Briefly, the accuracy of each new model was compared with that of the conventional TRISS model by the two Ps intervals (<0.50 and ≥0.5). Assessments were made using the TRISS methodology and four age groups (neonates/infants: 0 years, preschool children: 1–5 years, school children: 6–11 years, and adolescents: 12–18 years). Results of the analysis were shown as the medians or mean and interquartile ranges (IQRs; 25th–75th percentile) for continuous variables and as the numbers and percentages for categorical variables [5]. The analyses were performed by STATA/SE software, version 17.0 (StataCorp, College Station, TX, USA). Statistical differences were defined as a two-tailed *p* value < 0.05.

## 3. Results

A total of 17,738 pediatric patients with blunt trauma were included (Figure 1) and categorized by Ps-interval and age groups: Ps-interval < 0.50 (*n* = 17,399, 98%) and ≥0.5 (*n* = 339, 2%); neonates/infants (*n* = 330, 2%), preschool children (*n* = 2181, 12%), school children (*n* = 5947, 34%), and adolescents (*n* = 9280, 52%). The overall in-hospital mortality rate was 2.1%. Table 1 compares the predictor variables in derivation groups between the survivors and dead individuals. There were significant differences in all predictors of TRISS. Moreover, the median number of injury regions with AIS ≥ 3 per dead patient was higher than that of survivors (2 vs. 1, *p* < 0.001). The logistic regression models derived from the derivation dataset are shown with the conventional TRISS coefficients in Table 2. Among the 11 predictor models, Model 1 with RTS, ISS, and coded value of age year and Model 11 with ISS, age year, coded values of GCS, SBP, and RR, and the number of AIS ≥ 3 showed the best performance (AUC = 0.980, 95% CI = 0.973–0.987) (Table 2).

Table 3 and Table 4 showed demographics and predictor variables by two Ps-interval groups and four age categories. Table 5 showed the AUC of the TRISS model and the new predictor models by each Ps-interval group. The AUC of all models in patients with Ps < 0.5 was lower than that in patients with Ps ≥ 0.5. Model 11 showed the best performance (AUC = 0.762, 95% CI = 0.689–0.835) in patients with Ps < 0.5. Table 6 showed the AUC of the TRISS model and the new predictor models by age category. The AUC of all models in neonates/infants was lower than that in other age categories. In neonates/infants, Model 11 showed the best performance (AUC = 0.909, 95% CI = 0.972–0.988).

## 4. Discussion

The modified TRISS models were composed of different coefficients and variables from those used for the conventional TRISS model. We validated the new, modified models in Japanese pediatric patients with blunt trauma This study demonstrated that the new modified predictive Model 11 seemed to have the largest AUC value among all models. However, it is difficult to clarify that the new model had superior performance with statistical significance than the original TRISS model. Moreover, when considering the predictive performance of patients with Ps < 0.5, Model 11 and the original TRISS model were ineffective as prediction models because both had low AUC values of ˂0.8 and wide confidence intervals.
$$Ps=\frac{1}{\left(1+e^{-b}\right)}$$
$$b=-2.541-\left(0.085\times ISS\right)-\left(0.034\times AGE\right)+\left(1.161\times cGCS\right)+\left(0.925\times cSBP\right)+\;\left(0.434\times cRR\right)+\left(0.381\times No.of\;Injury\;Regions\;with\;AIS\geq 3\right)\;(\mathrm{Model}\;11)$$

The coefficients of the TRISS model were calculated based on the dataset from the Major Trauma Outcome Study [5]. Therefore, several previous studies have suggested that local and newly coefficients and predictors might further improve the accuracy and performance of the survival prediction model [10,11]. There is a steady yearly trend toward improved mortality in the JTDB dataset [15]; therefore, we attempted to estimate the coefficients derived from recent Japanese datasets. However, our results suggest that changing only the coefficients of the TRISS model did not contribute to the improvement of the predictive accuracy in Japanese pediatric patients.

Several factors may be responsible for this non-improvement of new models in pediatric patients with severe trauma, especially in severely injured patients and neonates/infants, as shown in the comparison of the TRISS model with Model 1 (Table 2, Table 5 and Table 6) as follows:

First, the lower proportion of patients aged <1 year (*n* = 330, 2% of all patients) and Ps-interval < 0.5 (*n* = 339, 2%) registered in the JTDB dataset than that in the other groups may be responsible for this non-improvement. Moreover, the aforementioned findings may also be due to the fact that younger patients accounted for most of the patients with missing data in the JTDB [6].

Second, age-related variation in physiological variables might yield a low accuracy in survival prediction models based on the TRISS methodology [16]. The cutoff values for coding RR, SBP, and age in the TRISS model were calculated using datasets derived mostly from adult patients; therefore, these values might not be associated with mortality among pediatric patients. Table 4 shows that 70% of neonates/infants had non-coded values of RR (=1), which were not normal, although an RR rate > 29 may be normal in neonates/infants. Moreover, a previous study suggested that the use of the actual age year as a continuous variable may contribute to the better prediction of Ps instead of the categorized variable as used in the TRISS model [10]. Our study demonstrated that a predictive model with age as a continuous variable had an equivalent AUC to that with age as a categorical variable, as shown in the comparison of Model 2 vs. Model 1 (0.980 vs. 0.980) in Table 2. This may be the reason why the mortality of injured pediatric patients in Japan differed by age groups (neonates/infants, 4.2%; preschool children, 2.0%; school children, 1.2%; and adolescents, 2.7%) and did not have a linear correlation with age, as shown in Table 4.

Third, pediatric trauma patients sustain the largest proportion of head injuries; evaluating the level of consciousness may be a more influential factor when predicting the survival outcome of pediatric patients with blunt trauma than that of adults [16]. In fact, our results demonstrated that the GCS scores of injured children who died were lower than those who survived (Table 1). Therefore, the prediction model with the GCS score as a predictor had a larger AUC value than that without the GCS score, as shown in the comparison of Model 3 vs. Model 6 (AUC 0.980 vs. 0.954) and Model 7 vs. Model 10 (0.976 vs. 0.931) in Table 2. On the other hand, evaluating physiological variables, especially the GCS score, is challenging due to age-related variations, such as limited verbal communication and motor response [17]. Moreover, younger pediatric patients had a larger rate of missing data on physiological variables, including the GCS score in the JTDB database [6]. There may be a possibility that neonates/infants and severely injured patients had a low performance of new predictive models in this study. As previous studies suggested, it may be effective to change the 12-point GCS score to a simplified predictor, such as the four-point AVPU scale: Alert, responsive to Verbal stimuli, responsive to Painful stimuli, and Unconscious [11].

Finally, as a consequence of constructing and validating the new model (Model 11) with the number of injury regions with AIS ≥ 3 as an additional variable because the proportion of patients with AIS ≥ 3 among dead individuals was higher than that among survivors, the model had the best performance for survival prediction in this study. However, the mortality of patients with severe anatomical injury and AIS ≥ 3, ISS ≥ 16, or ISS ≥ 18 has decreased from more than 20% to 9–12.3% according to improvements in the trauma care system [18]. For better predictive accuracy, factors which are most associated with the mortality of trauma patients than anatomical severity, such as acidosis or coagulation risk factors based on laboratory values, should be utilized as survival predictors [18,19]. Furthermore, previous studies described that pediatric trauma score (PTS) and shock index pediatric adjusted (SIPA) had a high performance in predicting outcomes by evaluating the age-adjusted physiological status [20,21,22]. Therefore, for better predictive accuracy, age-adjusted factors should also be utilized as survival predictors for children.

In summary, our results suggest that the development of the nationwide dataset with less data missing and laboratory data, setting up the age-related cutoff values for coding the physiological variables, and improving the evaluation accuracy of consciousness for younger pediatric patients with head injuries may be useful to enhance the prediction accuracy of the survival outcome in Japanese children with blunt trauma.

The survival probability can describe the severity of injuries or prognosis of injured patients using a single numerical value. By using score values, we can easily compare the survival outcome between injured patients with equal injury severity and the quality of trauma care in each facility, if patients treated in several facilities had different severity. Therefore, in the future, it is essential to develop a convenient prediction model based on the new predictive variables and cutoff values associated with the survival mortality of injured children in Japan.

There were several limitations in this study. First, selection bias is a possibility because not all Japanese hospitals that provide treatment have been included in the JTDB. A total of 2480 and 5421 pediatric injured patients who registered in the JTDB dataset had missing survival outcome data and TRISS predictors, respectively. Almost 30% of the total pediatric injured patients with incomplete data were excluded from our study. As this may have introduced selection bias, a high-quality nationwide dataset with fewer missing data should be constructed for the high-accuracy predicting model in children. Additionally, the number of registered facilities differed during the study period. Furthermore, the JTDB had been using the AIS 90 until 2018 and is now using the ‘AIS 2005 Update 2008’ coding scale. Therefore, in the future, there is a need to conduct similar studies utilizing the newest measure in order to ensure the quality of our results.

## 5. Conclusions

We found that the predictive ability of the newly modified models was not superior to that of the current TRISS model; moreover, our proposed model had some limitations for children with blunt trauma who are younger and/or had a higher severity. Therefore, in the future, it is essential to develop a highly accurate prediction model based on the new predictive variables and cutoff values associated with the mortality of injured Japanese children by using a nationwide dataset with fewer missing data and added variables that can evaluate the age-related physiological and anatomical severity of injured patients.

## Figures and Tables

**Figure 1 children-10-01542-f001:**
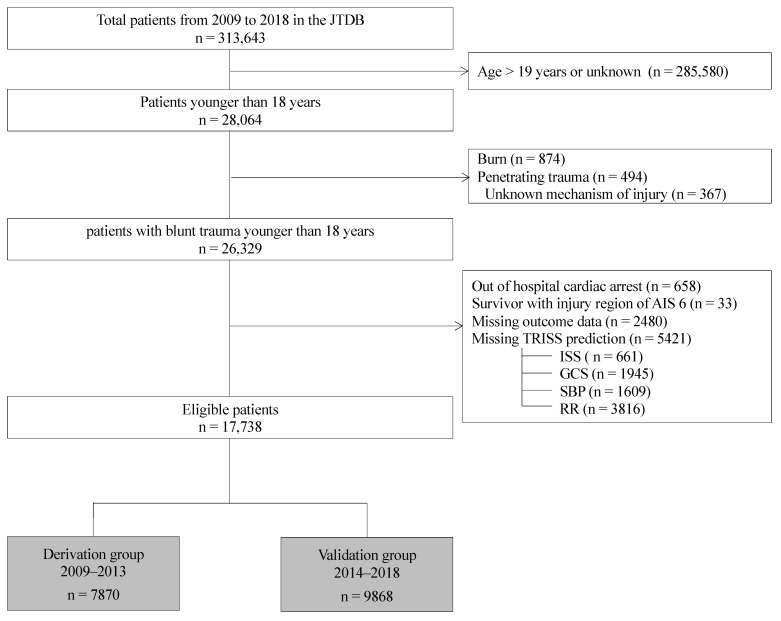
Flow diagram of the patient selection and inclusion process. AIS, Abbreviated Injury Scale; ISS, Injury Severity Score; GCS, Glasgow Coma Scale; SBP, systolic blood pressure; RR, respiratory rate.

**Table 1 children-10-01542-t001:** Comparison of demographics and predictor variables in the derivation groups between survivors and dead patients.

Variables	Coded Value	Survivor *n* = 7650	Dead *n* = 220	*p* Value
Age, years (median IQR)		13 (8–17)	16 (9–17)	<0.001
(mean ± SD)		11.9 ± 5.2	12.8 ± 5.6	
Male, *n* (%)		5499 (72)	154 (70)	0.541
Glasgow Coma Scale score, *n* (%)				
13–15	4	6252 (82)	14 (6)	<0.001
9–12	3	758 (10)	12 (5)	0.028
6–8	2	464 (6)	41 (19)	<0.001
4–5	1	73 (1)	40 (18)	<0.001
3	0	103 (1)	113 (51)	<0.001
Systolic blood pressure, mmHg, *n* (%)				
>89	4	7394 (97)	138 (63)	<0.001
76–89	3	157 (2)	14 (6)	<0.001
50–75	2	68 (0.9)	26 (12)	<0.001
1–49	1	31 (0.4)	42 (19)	<0.001
Respiratory rate, bpm, *n* (%)				
10–29	4	6310 (82)	111 (50)	<0.001
>29	3	1309 (17)	75 (34)	<0.001
6–9	2	21 (0.3)	3 (1)	0.004
1–5	1	3 (0.04)	4 (2)	<0.001
0	0	7 (0.1)	27 (12)	<0.001
Injury region, *n* (%)				
Head injury with AIS ≥ 3		2668 (35)	188 (85)	<0.001
Facial injury with AIS ≥ 3		69 (1)	5 (2)	0.038
Neck injury with AIS ≥ 3		7 (0.1)	1 (0.5)	0.096
Chest injury with AIS ≥ 3		1445 (19)	140 (64)	<0.001
Abdominal and pelvic injury with AIS ≥ 3		626 (8)	43 (20)	<0.001
Spinal injury with AIS ≥ 3		297 (4)	15 (7)	0.028
Upper extremity injury with AIS ≥ 3		678 (9)	7 (3)	0.003
Lower extremity injury with AIS ≥ 3		1182 (15)	53 (24)	0.001
Skin injury with AIS ≥ 3		0	0	-
Number of injury regions with AIS ≥ 3 per patients, (median IQR)		1 (0–1)	2 (1–3)	<0.001
Injury Severity Score, (median IQR)		10 (8–17)	34 (25–45)	<0.001
Revised Trauma Score, (median IQR)		7.84 (7.55–7.84)	4.09 (2.93–5.03)	<0.001
Survival probability, (median IQR)		0.99 (0.98–0.99)	0.48 (0.17–0.78)	<0.001
(mean ± SD)		0.97 ± 0.08	0.48 ± 0.31	
Probability of survival ≥ 0.50, *n* (%)		7589 (99)	108 (49)	<0.001
Probability of survival < 0.50, *n* (%)		61 (1)	112 (51)	<0.001

AIS, Abbreviated Injury Scale; IQR, interquartile range; SD, standard deviation. Table 1 compares the predictor variables in derivation groups between the survivors and dead individuals.

**Table 2 children-10-01542-t002:** Survival prediction models made for derivation groups, and results of validation analysis in all validation groups.

	Coefficient of Logistic Regression Models									AUC (95% CI)	
	Intercept	RTS	ISS	cAge	Age	cGCS	cSBP	cRR	No. of Injury Region with AIS ≥ 3	Overall (*n* = 9868)	*p* Value
TRISS model	−0.450	0.809	−0.084	−1.743	-	-	-	-	-	0.979 (0.972–0.986)	
Model 1	−2.448	1.235	−0.068	0.000	-	-	-	-	-	0.980 (0.973–0.987)	0.499
Model 2	−2.220	1.244	−0.065	-	−0.029	-	-	-	-	0.980 (0.973–0.987)	0.794
Model 3	−2.320	-	−0.065	-	−0.029	1.160	0.879	0.430	-	0.980 (0.973–0.987)	0.647
Model 4	−1.185	-	−0.066	-	−0.023	1.202	0.946	-	-	0.978 (0.970–0.986)	0.401
Model 5	0.406	-	−0.074	-	−0.015	1.164	-	0.592	-	0.979 (0.972–0.985)	0.531
Model 6	0.188	-	−0.114	-	−0.013	-	0.913	0.765	-	0.954 (0.940–0.968)	<0.001
Model 7	2.203	-	−0.075	-	0.000	1.232	-	-	-	0.976 (0.968–0.984)	0.014
Model 8	2.256	-	−0.120	-	0.005	-	1.066	-	-	0.944 (0.927–0.960)	<0.001
Model 9	2.899	-	−0.122	-	−0.004	-	-	0.970	-	0.942 (0.927–0.958)	<0.001
Model 10	6.237	-	−0.130	-	0.021	-	-	-	-	0.931 (0.913–0.948)	<0.001
Model 11	−2.541	-	−0.085	-	−0.034	1.161	0.925	0.434	0.381	0.980 (0.973–0.987)	0.667

TRISS, Trauma and Injury Severity Score; RTS, Revised Trauma Score; ISS, Injury Severity Score; cAge, coded value of age year; Age, age year; cGCS, coded value of Glasgow coma scale; cSBP, coded value of systolic blood pressure; cRR, coded value of respiratory rate; AIS, Abbreviated Injury Scale; AUC, area under the receiver operating characteristic; CI, confidence interval. The logistic regression models derived from the derivation dataset are shown with the coefficients of conventional model. Among the 11 predictor models, Model 1 showed the best performance.

**Table 3 children-10-01542-t003:** Demographics and predictor variables by two probability of the survival interval groups.

Variables	Patients with TRISS Ps ≥ 0.5 (*n* = 17,399)	Patients with TRISS Ps < 0.5 (*n* = 339)
Age, years (median IQR)	13 (8–16)	15 (9–17)
(mean ± SD)	11.8 ± 5.2	12.8 ± 5.4
Male, *n* (%)	12,663 (73)	237 (70)
Glasgow Coma Scale score, *n* (%)		
13–15	14,567 (84)	5 (1)
9–12	1516 (9)	11 (3)
6–8	939 (5)	47 (14)
4–5	159 (1)	44 (13)
3	218 (1)	232 (68)
Systolic blood pressure, mmHg, *n* (%)		
>89	16,880 (97)	172 (51)
76–89	342 (2)	35 (10)
50–75	108 (0.6)	60 (18)
1–49	69 (0.4)	72 (21)
Respiratory rate, /min, *n* (%)		
10–29	14,518 (83)	154 (45)
>29	2822 (16)	120 (35)
6–9	42 (0.2)	5 (1)
1–5	4 (0.02)	5 (1)
0	13 (0.1)	55 (16)
Injury region, *n* (%)		
Head injury with AIS ≥ 3	5716 (33)	290 (86)
Facial injury with AIS ≥ 3	159 (1)	14 (4)
Neck injury with AIS ≥ 3	15 (0.1)	5 (1)
Chest injury with AIS ≥ 3	3056 (18)	280 (83)
Abdominal and pelvic injury with AIS ≥ 3	1243 (7)	102 (30)
Spinal injury with AIS ≥ 3	711 (4)	29 (9)
Upper extremity injury with AIS ≥ 3	1589 (9)	21 (6)
Lower extremity injury with AIS ≥ 3	2603 (15)	130 (38)
Skin injury with AIS ≥ 3	3 (0.02)	0
Number of injury region with AIS ≥ 3 per patients, (median IQR)	1 (0–1)	3 (2–3)
Injury Severity Score, (median IQR)	9 (5–17)	43 (35–50)
Revised Trauma Score, (median IQR)	7.84 (7.55–7.84)	3.80 (2.63–4.09)
Survival probability, (median IQR)	0.99 (0.99–0.99)	0.27 (0.13–0.40)
(mean ± SD)	0.97 ± 0.06	0.26 ± 0.15
Actual mortality, *n* (%)	179 (1.0)	199 (58.7)

TRISS, Trauma and Injury Severity Score; AIS, Abbreviated Injury Scale; IQR, interquartile range; Ps, survival probability; SD, standard deviation. Table 3 show demographics and predictor variables by two Ps-interval groups.

**Table 4 children-10-01542-t004:** Demographics and predictor variables by four age categories.

Variables	Neonates/Infants (*n* = 330)	Pre-School Children (*n* = 2181)	School Children (*n* = 5947)	Adolescents (*n* = 9280)
Age, years (median IQR)	0 (0–0)	3 (2–4)	9 (7–11)	16 (15–17)
(mean ± SD)	0 ± 0	3.2 ± 1.4	8.9 ± 2.0	16.1 ± 1.6
Male, *n* (%)	219 (66)	1398 (64)	4261 (71)	7022 (76)
Glasgow Coma Scale score, *n* (%)				
13–15	112 (68)	1651 (76)	5088 (86)	7610 (82)
9–12	53 (16)	305 (14)	451 (8)	718 (8)
6–8	37 (11)	133 (6)	263 (4)	553 (6)
4–5	3(1)	29 (1)	46 (1)	125 (1)
3	14 (4)	63 (3)	99 (2)	274 (3)
Systolic blood pressure, mmHg, *n* (%)				
>89	260 (79)	2032 (93)	5798 (97)	8962 (97)
76–89	35 (11)	85 (4)	89 (2)	168 (2)
50–75	13 (4)	30 (1)	35 (0.6)	90 (1)
1–49	22 (7)	34 (2)	25 (0.4)	60 (0.7)
Respiratory rate, /min, *n* (%)				
10–29	96 (29)	1376 (63)	5027 (85)	8173 (88)
>29	232 (70)	791 (36)	893 (15)	1026 (11)
6–9	0	2 (0.1)	13 (0.2)	32 (0.3)
1–5	0	0	2. (0.03)	7 (0.1)
0	2 (0.6)	12 (0.6)	12 (0.2)	42 (0.5)
Injury region, *n* (%)				
Head injury with AIS ≥ 3	266 (81)	855 (39)	1985 (33)	2900 (31)
Facial injury with AIS ≥ 3	0	12 (0.6)	43 (0.7)	118 (1)
Neck injury with AIS ≥ 3	0	3 (0.1)	3 (0.1)	14 (0.2)
Chest injury with AIS ≥ 3	10 (3)	349 (16)	860 (15)	2117 (23)
Abdominal and pelvic injury with AIS ≥ 3	3 (1)	86 (4)	468 (8)	788 (8)
Spinal injury with AIS ≥ 3	1 (0.3)	25 (1)	84 (1)	630 (7)
Upper extremity injury with AIS ≥ 3	0	213 (10)	846 (14)	551 (6)
Lower extremity injury with AIS ≥ 3	13 (4)	233 (11)	828 (14)	1659 (18)
Skin injury with AIS ≥ 3	0	1 (0.1)	1 (0.02)	1 (0.01)
Number of injury region with AIS ≥ 3 per patients, (median IQR)	1 (1–1)	1 (0–1)	1 (0–1)	1 (0–1)
Injury Severity Score, (median IQR)	16 (9–17)	9 (5–16)	9 (8–16)	10 (6–19)
Revised Trauma Score, (median IQR)	7.55 (6.61–7.55)	7.55 (6.90–7.84)	7.84 (7.55–7.84)	7.84 (7.55–7.84)
Survival probability, (median IQR)	0.99 (0.95–0.99)	0.99 (0.98–0.99)	0.99 (0.99–0.99)	0.99 (0.98–0.99)
(mean ± SD)	0.94 ± 0.13	0.96 ± 0.11	0.97 ± 0.09	0.95 ± 0.13
Probability of survival ≥ 0.50, *n* (%)	321 (97)	2141 (98)	5884 (99)	9053 (98)
Probability of survival < 0.50, *n* (%)	9 (3)	40 (2)	63 (1)	227 (2)
Actual mortality, *n* (%)	14 (4.2)	43 (2.0)	73 (1.2)	248 (2.7)

AIS, Abbreviated Injury Scale; IQR, interquartile range; SD, standard deviation. Table 4 show demographics and predictor variables by four age categories.

**Table 5 children-10-01542-t005:** AUC of TRISS and new models for survival probabilities <0.5 and ≥0.5 in the validation groups.

	AUC (95% CI)			
	Patients with TRISS Ps ≥ 0.5 (*n* = 9702)	*p* Value	Patients with TRISS Ps < 0.5 (*n* = 166)	*p* Value
TRISS model	0.965 (0.951–0.979)		0.718 (0.642–0.795)	
Model 1	0.966 (0.953–0.979)	0.538	0.736 (0.660–0.812)	0.460
Model 2	0.965 (0.951–0.979)	0.853	0.747 (0.673–0.822)	0.237
Model 3	0.966 (0.952–0.979)	0.709	0.750 (0.676–0.825)	0.203
Model 4	0.962 (0.945–0.979)	0.451	0.709 (0.631–0.787)	0.717
Model 5	0.965 (0.953–0.977)	0.837	0.674 (0.591–0.757)	0.210
Model 6	0.912 (0.886–0.938)	<0.001	0.702 (0.623–0.781)	0.655
Model 7	0.960 (0.945–0.976)	0.147	0.588 (0.539–0.710)	0.001
Model 8	0.894 (0.862–0.927)	<0.001	0.625 (0.539–0.710)	0.017
Model 9	0.900 (0.871–0.929)	<0.001	0.654 (0.570–0.739)	0.152
Model 10	0.887 (0.854–0.919)	<0.001	0.509 (0.419–0.598)	<0.001
Model 11	0.966 (0.952–0.979)	0.757	0.762 (0.689–0.835)	0.095

TRISS, Trauma and Injury Severity Score; AUC, area under the receiver operating characteristic; CI, confidence interval. The AUC of all models in patients with Ps < 0.5 was lower than that in patients with Ps ≥ 0.5. Model 11 showed the best AUC in patients with Ps < 0.5.

**Table 6 children-10-01542-t006:** AUC of TRISS and new models by age groups in validation groups.

	AUC (95% CI)							
	Neonates/Infants (*n* = 197)	*p* Value	Preschool Children (*n* = 1260)	*p* Value	School Children (*n* = 3334)	*p* Value	Adolescents (*n* = 5077)	*p* Value
TRISS model	0.905 (0.762–1.000)		0.982 (0.972–0.991)		0.983 (0.967–0.999)		0.979 (0.970–0.988)	
Model 1	0.908 (0.757–1.000)	0.763	0.983 (0.974–0.992)	0.133	0.981 (0.964–0.997)	0.157	0.980 (0.972–0.988)	0.331
Model 2	0.908 (0.757–1.000)	0.763	0.983 (0.974–0.992)	0.170	0.980 (0.964–0.997)	0.161	0.980 (0.973–0.988)	0.290
Model 3	0.908 (0.757–1.000)	0.763	0.983 (0.974–0.992)	0.137	0.981 (0.965–0.997)	0.266	0.980 (0.973–0.988)	0.279
Model 4	0.903 (0.744–1.000)	0.704	0.982 (0.972–0.991)	0.932	0.977 (0.955–0.999)	0.063	0.980 (0.972–0.987)	0.523
Model 5	0.905 (0.786–1.000)	0.940	0.982 (0.974–0.991)	0.760	0.980 (0.965–0.995)	0.276	0.978 (0.970–0.986)	0.749
Model 6	0.895 (0.770–1.000)	0.665	0.921 (0.870–0.972)	0.011	0.976 (0.961–0.991)	0.294	0.953 (0.934–0.971)	<0.001
Model 7	0.899 (0.769–1.000)	0.517	0.980 (0.971–0.989)	0.482	0.976 (0.954–0.998)	0.048	0.977 (0.969–0.985)	0.257
Model 8	0.881 (0.736–1.000)	0.403	0.923 (0.880–0.966)	0.002	0.974 (0.952–0.996)	0.076	0.938 (0.916–0.961)	<0.001
Model 9	0.797 (0.684–0.910)	0.103	0.916 (0.863–0.968)	0.008	0.968 (0.952–0.984)	0.068	0.943 (0.923–0.963)	<0.001
Model 10	0.787 (0.644–0.929)	0.096	0.930 (0.891–0.969)	0.003	0.968 (0.946–0.989)	0.010	0.923 (0.899–0.946)	<0.001
Model 11	0.909 (0.759–1.000)	0.607	0.985 (0.976–0.993)	0.005	0.981 (0.966–0.996)	0.341	0.980 (0.972–0.988)	0.448

TRISS, Trauma and Injury Severity Score; AUC, area under the receiver operating characteristic; CI, confidence interval. The AUC of all models in neonates/infants was lower than that in other age categories. In neonates/infants, Model 11 showed the best AUC.

## Data Availability

The datasets generated during and analyzed during the current study are not publicly available due to data access control system for limiting data access that the Japan Association for the Surgery of Trauma determines whether to permit data usage in each application but are available from the corresponding author on reasonable request.

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
