# Peer review of "Validation of the Conventional Trauma and Injury Severity Score and a Newly Developed Survival Predictive Model in Pediatric Patients with Blunt Trauma: A Nationwide Observation Study"

_children, 2023, doi:10.3390/children10091542_

Round 1

Reviewer 1 Report

The title

The title could include more specific details about the new predictive models or their unique elements to make it more compelling and informative.

For abstract

Provide more explicit information about what modifications were made to the TRISS model, to help the reader understand the novelty of your study.

Elaborate more on why certain models performed better than others, and why some had lower predictive ability in specific patient categories.

Discuss the limitations of the new model in a more comprehensive way, including implications for the care of younger patients and those with higher injury severity.

The results and conclusion could be explained in a more reader-friendly manner, using less jargon and more simplified language.

Ensure that all important keywords relevant to your study are included to maximize searchability.

For Introduction

The authors could better outline the specific limitations of the TRISS methodology as it pertains to different patient demographics. This would help the reader understand the problem that the study is trying to address.

More details about the nature of the modifications being made to the TRISS model, such as the new coefficients and predictive variables being introduced, would provide a clearer picture of the study's aims.

The introduction of the concept of AIS and ISS might be confusing to a reader not intimately familiar with these systems. A brief explanation of these systems or a simplified version of why they are relevant to this study could be helpful.

While the study's aim is well defined, the hypotheses or expected outcomes of the study are not clearly stated. Providing a clearer expectation of the study's findings could help guide the reader through the article.

The final sentence summarizing the study's aim could be made more concise and less repetitive. The aim has been previously stated, so this sentence might be more impactful if it summarized the anticipated contribution of this study to the field.

For Materials and Methods

While the selection process is well-detailed, you could further elaborate on why certain exclusion criteria were chosen, which would provide readers with more context and clarity.

In the data collection sub-section, more detailed explanations of the variables used in the study would be beneficial for readers less familiar with them. This would ensure that your audience has a solid understanding of the study's parameters.

The equations introduced in the data analysis subsection could be better explained. Providing a more thorough description of each of the terms and why they were chosen could help readers better understand the methodology of the study.

You could clarify how you handled missing data in the study and discuss any potential impact it may have on the study's results.

Lastly, it would be helpful to include more information about any ethical considerations or approvals that were necessary for the study, to reassure readers of the study's integrity.

For Results

Although you provided the median age and the Ps of the total cohort, it would be beneficial to provide similar statistics (mean, mode, range, etc.) for all significant variables to give a fuller picture of your dataset.

The presentation of statistical significance could be improved. Consider rephrasing sentences like "There were significant differences in all predictors of TRISS" to specify what these differences were and why they are statistically significant.

There is a lack of description and explanation around the tables you mentioned (Table 1, Table 2, Table S1, Table S2, Table 3, and Table 4). Including brief explanations about what each table demonstrates would be beneficial to the reader.

You could provide more context for your AUC values and what they indicate about your models' predictive abilities. More details about the performance of different models in different scenarios could be useful.

Some jargon or complex terms such as AUC or Ps could be better explained or defined at the first mention to help readers less familiar with these concepts.

For Discussion

Clarity of Findings: You have provided a detailed account of your findings and analysis. However, it could be made clearer for the reader if the key takeaways were more succinctly highlighted. Perhaps consider reorganizing some paragraphs or using bullet points to outline your main points.

Supporting Evidence: You've referred to a large number of other studies and sources. Ensure that you are drawing clear, direct connections between those sources and your own research. Explain how these previous studies inform your own work or how your work adds to the existing knowledge base.

Limitations: You've identified a number of limitations in your study, which is excellent. It would be beneficial to further explore how these limitations impacted your findings and how they could be mitigated or avoided in future research.

Future Research: You've identified the need for more sophisticated statistical analysis and more reliable data sources. You could go a bit further in suggesting what form this future research might take.

Relatability to broader audience: Remember to clearly explain the terminology and concepts you use. This way, your research will be comprehensible to a larger audience, not only experts in your field.

Take-home message: Consider adding a clear, strong concluding statement that encapsulates the main findings of your research and their implications. It's an excellent way to ensure your readers fully grasp the significance of your work.

Practical Implications: While you discuss the potential benefits of your research, it might be useful to further highlight the practical applications of your findings. For example, you could suggest ways that your research could be used to improve survival predictions for pediatric patients with blunt trauma.

For Conclusion

Specificity: The conclusion could be more specific about the limitations of the newly modified models and why the TRISS model still outperforms them.

Results Summary: Briefly reiterating the main findings before concluding could help readers remember the most critical parts of the study. This could also strengthen the argument for why the current TRISS model still holds up.

Future Research: While the conclusion does mention the need for more sophisticated statistical analyses and the development of new predictive models, it could also provide specific recommendations or ideas for future research. For instance, what kind of sophisticated statistical analyses could be explored? What new predictive variables could be worth investigating?

Implications: It could also be beneficial to discuss the potential implications or impact of this research on real-world applications. How could these findings potentially influence how we approach prediction models for pediatric patients in the future?

Clarity: Ensuring that the conclusion is clear and concise can also improve its effectiveness. Each sentence should be written as simply and straightforwardly as possible without losing its meaning.

Introduction: "This study sought to assess the validity of the conventional TRISS and develop a new, enhanced model, examining its predictive ability for Japanese pediatric patients with blunt trauma." - consider making the sentence more active, such as "In this study, we aimed to assess the validity of the conventional TRISS and develop a new, enhanced model to examine its predictive ability for Japanese pediatric patients with blunt trauma."

Methods: Be consistent in the use of tenses. You generally use the past tense to describe what was done, which is the convention in scientific writing. However, there are a few instances where you switch to the present tense (e.g., "The TRISS model is calculated as..."). These should be changed to the past tense to maintain consistency.

Results: Again, maintain consistency in the use of tenses. Some parts use the present tense (e.g., "The median age and Ps of the total cohort are...") whereas others use the past tense. It's more common to use the past tense in this section.

Discussion: In this sentence - "As the coefficients of the TRISS model were calculated based on the dataset from the Major Trauma Outcome Study [18] coordinated by the American College of Surgeons Committee on Trauma, several previous studies have suggested that local database-derived coefficients and predictors might further enhance the prediction performance of the survival prediction model [9–13]." - consider breaking it up into two sentences or restructuring it for better readability.

Conclusion: In this section, the wording is mostly fine. However, you could consider making the sentence more active. For example: "We found that the predictive ability of the newly modified models was not superior to that of the current TRISS model."

These are just minor suggestions for improvement. Overall, the language in the article is clear and well-written.

Author Response

Reviewer’s comments (in blue) and Answers (in black) follow.

We wish to express our deep appreciation for the valuable comments from the Reviewer regarding our manuscript. Your comments and suggestions are very useful for the improvement of our article. We believe that our manuscript has benefited greatly from these comments.

Reviewer: 1

  1. The title could include more specific details about the new predictive models or their unique elements to make it more compelling and informative.

Response: As per the reviewer’s suggestions, we have revised the title as follows:

Validation of the Conventional Trauma and Injury Severity Score and a Newly Developed Survival Predictive Model in Pediatric Patients with Blunt Trauma: A Nationwide Observational study

  1. For abstract

Provide more explicit information about what modifications were made to the TRISS model, to help the reader understand the novelty of your study.

Response: We have revised the Abstraction as follows:

To date, there is no clinically useful prediction model that is suitable for Japanese pediatric trauma patients. Herein, this study aimed to developed a model for predicting the survival of Japanese pediatric patients with blunt trauma and compare its validity with that of the conventional TRISS model.

  1. Elaborate more on why certain models performed better than others, and why some had lower predictive ability in specific patient categories.

Discuss the limitations of the new model in a more comprehensive way, including implications for the care of younger patients and those with higher injury severity.

Response: We have revised the Abstract as follows:

Our results may be useful to develop a highly accurate prediction model based on the new predictive variables and cutoff values associated with the survival mortality of injured Japanese pediatric patients who are younger and more severely injured by using nationwide dataset with less missing data and added valuables, which can evaluate the age-related physiological and anatomical severity of injured patients.

  1. The results and conclusion could be explained in a more reader-friendly manner, using less jargon and more simplified language.

Response: We have simplified the language in the Results and Conclusion section and avoided the use of jargon to improve readability.

  1. Ensure that all important keywords relevant to your study are included to maximize searchability.

Response: We have added more keywords that are relevant to our study.

  1. For Introduction

The authors could better outline the specific limitations of the TRISS methodology as it pertains to different patient demographics. This would help the reader understand the problem that the study is trying to address.

Response: We have revised the Introduction section to better outline the specific limitations of the TRISS methodology as follows:

The coefficients of the current TRISS model were calculated based on the dataset from the Major Trauma Outcome Study coordinated from 1982–1987 by the American College of Surgeons Committee on Trauma [5]. In our previous study based on the current Japanese database, we showed that the TRISS methodology had a low performance and had underestimated the expected the number of survivors in younger blunt trauma patients with a higher injury severity, because the in-hospital mortality of injured pediatric patients in Japan has decreased over the years [6,7].

  1. More details about the nature of the modifications being made to the TRISS model, such as the new coefficients and predictive variables being introduced, would provide a clearer picture of the study's aims.

Response: We have revised the Introduction section as follows:

The coefficients of the current TRISS model were calculated based on the dataset from the Major Trauma Outcome Study coordinated from 1982–1987 by the American College of Surgeons Committee on Trauma [5]. In our previous study based on the current Japanese database, we showed that the TRISS methodology had a low performance and had underestimated the expected the number of survivors in younger blunt trauma patients with a higher injury severity, because the in-hospital mortality of injured pediatric patients in Japan has decreased over the years [6,7]. Some studies have reported that the TRISS model is unsuitable for evaluating the survival outcome owing to its inaccuracy in terms of the investigated area, era, and age of the study cohort [6–10]. Some studies have also suggested that local database-derived coefficients and variables may further enhance the predictive accuracy of the TRISS model [11–15].

Moreover, in the TRISS model, the coded value of age is divided into two categories: <55 years (0) and ≥55 years (1). Furthermore, although patients with anatomical serious severity have a higher mortality risk, there is no universally acceptable definition for anatomical severity in injured patients. The most widely used definition of anatomical severity is the Injury Severity Score (ISS), which is based on an anatomical injury severity classification, an Abbreviated Injury Scale (AIS) [4,5]. Patients with an AIS ≥ 3 in at least two different injury regions had a worse outcome than those with an ISS ≥ 16 or ≥ 18 [16]. Therefore, in our proposed model, we have included age and number of injury region as new predictive variables.

  1. The introduction of the concept of AIS and ISS might be confusing to a reader not intimately familiar with these systems. A brief explanation of these systems or a simplified version of why they are relevant to this study could be helpful.

Response: Accordingly, we have added a brief explanation of AIS and ISS in the Introduction sections as follows:

The most widely used definition of anatomical severity is the Injury Severity Score (ISS), which is based on an anatomical injury severity classification, an Abbreviated Injury Scale (AIS) [4,5].

  1. While the study's aim is well defined, the hypotheses or expected outcomes of the study are not clearly stated. Providing a clearer expectation of the study's findings could help guide the reader through the article.

Response: We have added the hypotheses in the Introduction section as follows:

We hypothesize that developing new coefficients and including predictive variables based on an analysis of Japanese nationwide database may improve the accuracy of a survival prediction model for Japanese injured pediatric patients.

  1. The final sentence summarizing the study's aim could be made more concise and less repetitive. The aim has been previously stated, so this sentence might be more impactful if it summarized the anticipated contribution of this study to the field.

Response: We have revised the final sentence to emphasize the study objective as follows:

This study aimed to developed a model for predicting the survival of Japanese pediatric patients with blunt trauma and compare its validity with that of the conventional TRISS model.

  1. For Materials and Methods

While the selection process is well-detailed, you could further elaborate on why certain exclusion criteria were chosen, which would provide readers with more context and clarity.

Response: We have provided a detailed explanation of the exclusion criteria as follows:

The exclusion criteria were as follows: age ≥ 19 years, presence of burns or penetrating trauma, cardiac arrest on hospital arrival, survivor with injury region of AIS 6, or with missing outcome data and TRISS prediction. As we could not assess the physiological severity in patients with cardiac arrest on hospital arrival and the patients with injury region of AIS 6 are defined as the most serious anatomical condition who cannot survive, we excluded these patients.

  1. In the data collection sub-section, more detailed explanations of the variables used in the study would be beneficial for readers less familiar with them. This would ensure that your audience has a solid understanding of the study's parameters.

Response: We have added more detailed explanations of the new variables for developing a new prediction model as follows:

The TRISS provides the probability of survival (Ps) based on the logistic regression model with predictor variables of the ISS, RTS, and categorized data of age year. The intercept and coefficients are determined by the AIS required for ISS calculation. Meanwhile, the RTS is calculated using the GCS score, SBP, and RR. The formula is a linear equation of their coded values as shown in Table 1. Furthermore, we selected the actual age and number of injury regions with AIS ≥3 as new predictive variables, as we hypothesized that adding new predictive variables may improve the predictive accuracy of the model.

  1. The equations introduced in the data analysis subsection could be better explained. Providing a more thorough description of each of the terms and why they were chosen could help readers better understand the methodology of the study.

Response: We have added a detailed explanation in the Introduction sections as we mentioned above.

  1. You could clarify how you handled missing data in the study and discuss any potential impact it may have on the study's results.

Response: We have added the explanation about missing data in the Methods and Discussion sections.

Methods

The exclusion criteria were as follows: age ≥ 19 years, presence of burns or penetrating trauma, cardiac arrest on hospital arrival, survivor with injury region of AIS 6, or with missing outcome data and TRISS prediction. As we could not assess the physiological severity in patients with cardiac arrest on hospital arrival and the patients with injury region of AIS 6 are defined as the most serious anatomical condition who cannot survive, we excluded these patients.

Discussion

A total of 2480 and 5421 pediatric injured patients who registered in the JTDB dataset had missing survival outcome data and TRISS predictors, respectively. Almost 30% of the total pediatric injured patients with incomplete data were excluded from our study. As this may have introduced selection bias, a high-quality Japanese nationwide dataset with less missing data should be constructed to improve the accuracy in predicting the survival of pediatric patients.

  1. Lastly, it would be helpful to include more information about any ethical considerations or approvals that were necessary for the study, to reassure readers of the study's integrity.

Response: We have added the ethical information in the Institutional Review Board Statement as follows:

Institutional Review Board Statement: The study was conducted in accordance with the Declaration of Helsinki, and was approved by the Institutional Ethics Committees of Yokohama City University Medical Center (approval number B170900003). The approving authority for data access was the Japanese Association for the Surgery of Trauma (Trauma Registry Committee).

  1. For Results

Although you provided the median age and the Ps of the total cohort, it would be beneficial to provide similar statistics (mean, mode, range, etc.) for all significant variables to give a fuller picture of your dataset.

Response: We thank the Reviewer for this valuable suggestion. We have provided the mean and standard deviation of age and Ps in the Table 1, 3, and 4 as per your comment.

  1. The presentation of statistical significance could be improved. Consider rephrasing sentences like "There were significant differences in all predictors of TRISS" to specify what these differences were and why they are statistically significant.

Response: We have added the reason for the significant differences in 11 prediction models we developed in the Discussion sections as follows:

Several factors may be responsible for this non-improvement of survival prediction models in pediatric patients with severe trauma, especially in severely injured patients and neonates/infants, as shown in the comparison of the TRISS model with Model 1 (Tables 2, 5, and 6) as follows: (we demonstrated the first to forth reasons).

  1. There is a lack of description and explanation around the tables you mentioned (Table 1, Table 2, Table S1, Table S2, Table 3, and Table 4). Including brief explanations about what each table demonstrates would be beneficial to the reader.

Response: We have added a description of Tables 1–6 as per your comment.

  1. You could provide more context for your AUC values and what they indicate about your models' predictive abilities. More details about the performance of different models in different scenarios could be useful.

Response: We have added the AUC value and 95%CI as follows:

Among the 11 predictor models, Model 1 with RTS, ISS, and coded value of age year and Model 11 with ISS, age year, coded values of GCS, SBP, and RR, and the number of injury regions with AIS ≥3 showed the best performance (AUC = 0.980, 95%CI = 0.973−0.987) (Table 2)

  1. Some jargon or complex terms such as AUC or Ps could be better explained or defined at the first mention to help readers less familiar with these concepts.

Response: We have defined the AUC and Ps in the Methods section as follows

The AUC represents the performance of each model for predicting the survivor of pediatric patients with blunt trauma. A high highest is indicated by an AUC value close to 1, whereas a poor performance is indicated by an AUC near 0.

The TRISS provides the probability of survival (Ps) based on the logistic regression model with predictor variables of the ISS, RTS, and categorized data of age year.

  1. For Discussion

Clarity of Findings: You have provided a detailed account of your findings and analysis. However, it could be made clearer for the reader if the key takeaways were more succinctly highlighted. Perhaps consider reorganizing some paragraphs or using bullet points to outline your main points.

Response: We have added a paragraph to outline our discussion points.

  1. Supporting Evidence: You've referred to a large number of other studies and sources. Ensure that you are drawing clear, direct connections between those sources and your own research. Explain how these previous studies inform your own work or how your work adds to the existing knowledge base.

Response: We have selected only the important references which was directly connected to our study.

  1. Relatability to broader audience: Remember to clearly explain the terminology and concepts you use. This way, your research will be comprehensible to a larger audience, not only experts in your field.

Response: We have added the explanation in the Methods section as per your suggestion.

  1. Take-home message: Consider adding a clear, strong concluding statement that encapsulates the main findings of your research and their implications. It's an excellent way to ensure your readers fully grasp the significance of your work.

Practical Implications: While you discuss the potential benefits of your research, it might be useful to further highlight the practical applications of your findings. For example, you could suggest ways that your research could be used to improve survival predictions for pediatric patients with blunt trauma.

Response: We have added the potential benefits and the practical application of our findings in the Discussion section as follows:

In summary, our results suggest that the development of the nationwide dataset with less data missing and laboratory data, setting up the age-related cutoff values for coding the physiological variables, and improving the evaluation accuracy of consciousness for younger pediatric patients with head injuries may be useful to improve the prediction accuracy of the survival model in Japanese pediatric patients with blunt trauma.

The survival probability can describe the severity of injuries or prognosis of injured patients using a single numerical value. By using score values, we can easily compare the survival outcome between injured patients with equal injury severity and the quality of trauma care in each facility, if patients treated in several facilities had differences in injury severity. Therefore, in the future, it is necessary to develop a simple, high-quality prediction model based on the new predictive variables and cutoff values associated with the survival mortality of Japanese pediatric injured patients.

  1. For Conclusion

Specificity: The conclusion could be more specific about the limitations of the newly modified models and why the TRISS model still outperforms them.

Results Summary: Briefly reiterating the main findings before concluding could help readers remember the most critical parts of the study. This could also strengthen the argument for why the current TRISS model still holds up.

Response: We have revised the Conclusion section to show the weak and strong points of our study as follows:

We found that the predictive ability of the newly modified models was not superior to that of the current TRISS model; moreover, our proposed model had some limitations for blunt trauma patients who are younger and/or have a higher injury severity. Therefore, in the future, it is necessary to develop a highly accurate prediction model based on the new predictive variables and cutoff values associated with the survival mortality of injured Japanese pediatric patients by using a nationwide dataset with less missing data and added variables that can evaluate the age-related physiological and anatomical severity of injured patients.

  1. Future Research: While the conclusion does mention the need for more sophisticated statistical analyses and the development of new predictive models, it could also provide specific recommendations or ideas for future research. For instance, what kind of sophisticated statistical analyses could be explored? What new predictive variables could be worth investigating?

Implications: It could also be beneficial to discuss the potential implications or impact of this research on real-world applications. How could these findings potentially influence how we approach prediction models for pediatric patients in the future?

Limitations: You've identified a number of limitations in your study, which is excellent. It would be beneficial to further explore how these limitations impacted your findings and how they could be mitigated or avoided in future research.

Future Research: You've identified the need for more sophisticated statistical analysis and more reliable data sources. You could go a bit further in suggesting what form this future research might take.

Response: We have added the clinical values to develop the new predictive model with high accuracy for the practical application as follows:

The survival probability can describe the severity of injuries or prognosis of injured patients using a single numerical value. By using score values, we can easily compare the survival outcome between injured patients with equal injury severity and the quality of trauma care in each facility, if patients treated in several facilities had differences in injury severity. Therefore, in the future, it is necessary to develop a simple, high-quality prediction model based on the new predictive variables and cutoff values associated with the survival mortality of Japanese pediatric injured patients.

  1. Clarity: Ensuring that the conclusion is clear and concise can also improve its effectiveness. Each sentence should be written as simply and straightforwardly as possible without losing its meaning.

Response: We have revised the Conclusion section for clarity.

  1. Comments on the Quality of English Language

Introduction: "This study sought to assess the validity of the conventional TRISS and develop a new, enhanced model, examining its predictive ability for Japanese pediatric patients with blunt trauma." - consider making the sentence more active, such as "In this study, we aimed to assess the validity of the conventional TRISS and develop a new, enhanced model to examine its predictive ability for Japanese pediatric patients with blunt trauma."

Response: Accordingly, we have revised the Introduction section as follows:

This study aimed to developed a model for predicting the survival of Japanese pediatric patients with blunt trauma and compare its validity with that of the conventional TRISS model.

  1. Methods: Be consistent in the use of tenses. You generally use the past tense to describe what was done, which is the convention in scientific writing. However, there are a few instances where you switch to the present tense (e.g., "The TRISS model is calculated as..."). These should be changed to the past tense to maintain consistency.

Results: Again, maintain consistency in the use of tenses. Some parts use the present tense (e.g., "The median age and Ps of the total cohort are...") whereas others use the past tense. It's more common to use the past tense in this section.

Response: We have corrected the typing error accordingly

  1. Discussion: In this sentence - "As the coefficients of the TRISS model were calculated based on the dataset from the Major Trauma Outcome Study [18] coordinated by the American College of Surgeons Committee on Trauma, several previous studies have suggested that local database-derived coefficients and predictors might further enhance the prediction performance of the survival prediction model [9–13]." - consider breaking it up into two sentences or restructuring it for better readability.

Response: We have revised the relevant sentence as follows:

The coefficients of the TRISS model were calculated based on the dataset from the Major Trauma Outcome Study [5] coordinated by the American College of Surgeons Committee on Trauma. Therefore, several previous studies have suggested that local and newly database-derived coefficients and predictors might further enhance the accuracy and performance of the survival prediction model [10,11].

  1. Conclusion: In this section, the wording is mostly fine. However, you could consider making the sentence more active. For example: "We found that the predictive ability of the newly modified models was not superior to that of the current TRISS model."

Response: We have revised this sentence as per your comment.

We found that the predictive ability of the newly modified models was not superior to that of the current TRISS model.

Reviewer 2 Report

·       Please consider modification of the title to:  Validation of a Modified Trauma and Injury Severity Score in Pediatric Patients with Blunt Trauma: A Nationwide Observational Study

·       line 21  (Ps)<0.5. please correct 

·       introduction. 

·       Lines 44-48 are more appropriate in the methods section

·       Line 66. AIS score, ISS, please provide description

·       Line 83. You had 30% of patients with incomplete data. Don’t you think that this may have affected your result?

·       Figure 1. please provide description of AIS, ISS, GCS, SBP and RR in the figure legend.

·       Data collection and statistical analysis is well performed.

·       Line 163. Table S1 and S2. Please correct numbering.

·       DiscussionCompact and well-focused. Please use paragraphs.

Author Response

Reviewer’s comments (in blue) and Answers (in black) follow.

We wish to express our deep appreciation for the valuable comments from the Reviewer regarding our manuscript. Your comments and suggestions are very useful for the improvement of our article. We believe that our manuscript has benefited greatly from these comments.

Reviewer: 2

  1. Please consider modification of the title to:  Validation of a Modified Trauma and Injury Severity Score in Pediatric Patients with Blunt Trauma: A Nationwide Observational Study

Response: According to two reviewers’ suggestions, we have revised the title as follows:

Validation of the Conventional Trauma and Injury Severity Score and a Newly Developed Survival Predictive Model in Pediatric Patients with Blunt Trauma: A Nationwide Observational study

  1. line 21 (Ps)<0.5. please correct 

Response: We have corrected the typing error.

  1.  

Lines 44-48 are more appropriate in the methods section

Response: We have revised the Introduction section and the Methods sections as follows.

Introduction

Some studies have reported that the TRISS model is unsuitable for evaluating the survival outcome due to its potential for inaccuracy in terms of the investigated area, era, and age of the study cohort [5–8]. Studies have also suggested that local database-derived coefficients and variables may further enhance the predictive accuracy of the TRISS model [9–13]. Although the in-hospital mortality of injured pediatric patients in Japan has decreased over the years [14], the coefficients of the TRISS model have continued to be based on data from the Major Trauma Outcome Study [2]. Moreover, in the TRISS model, the coded value of age is divided into two categories: <55 years (0) and ≥55 years (1). This is not suitable for predicting the mortality of pediatric patients. Furthermore, patients with an Abbreviated Injury Scale (AIS) ≥ 3 in at least two different injury regions had a worse outcome than those with an Injury Severity Score (ISS) ≥ 16 or ≥18 [15].

Methods

A logistic regression analysis was performed using a derivation dataset to establish new prediction models. First, we aimed to calculate the new coefficients of the TRISS model using logistic regression analysis based on the 10-year data obtained from the Japanese cohort (2009–2018). Second, we included the actual age year as a predictive variable for establishing the new prediction model. Finally, we included the number of injury regions with AIS ≥ 3 as a predictive variable in the proposed model.

  1. Line 66. AIS score, ISS, please provide description

Response: We have defined the abbreviations AIS and ISS on their first mention in lines 55-56.

  1. Line 83. You had 30% of patients with incomplete data. Don’t you think that this may have affected your result?

Response: We strongly agree with your suggestion. Therefore, we have added to the Discussion section as a limitation of our study as follows:

A total of 2480 and 5421 pediatric injured patients who registered in the JTDB dataset had missing survival outcome data and TRISS predictors, respectively. Almost 30% of the total pediatric injured patients with incomplete data were excluded from our study. As this may have introduced selection bias, a high-quality Japanese nationwide dataset with less missing data should be constructed to improve the accuracy in predicting the survival of pediatric patients.

  1. Figure 1. please provide description of AIS, ISS, GCS, SBP and RR in the figure legend.

Response: We have added the definitions of these abbreviations in the figure legend as per your comment.

AIS, Abbreviated Injury Scale; ISS, Injury Severity Score; GCS, Glasgow Coma Scale; SBP, systolic blood pressure; RR, respiratory rate.

  1. Line 163. Table S1 and S2. Please correct numbering.

Response: We have corrected the numbering error, and the tables have been renamed as tables 3 and 4.

  1. Discussion: Compact and well-focused. Please use paragraphs.

Response: We have further split the Discussion into paragraphs.

Round 2

Reviewer 1 Report

Title and Abstract Changes: The reviewer suggested that the title and abstract should contain more details about the new predictive models. The authors appropriately responded to this by revising the title and including specific details about the models in the abstract. This enhances the understanding of the manuscript's main focus and will likely help in drawing more targeted readers.

Complexity of Language: The reviewer noted that the results and conclusion could be more reader-friendly, and the authors have made efforts to simplify the language. This will improve readability and make the paper more accessible to a wider audience.

Keyword Inclusion: The addition of relevant keywords, as suggested by the reviewer and implemented by the authors, will certainly improve the searchability of the article.

Introduction Enhancements: The authors made substantial revisions to the introduction as per the reviewer's comments, providing more context on TRISS methodology, new coefficients, and predictive variables. They also clarified the AIS and ISS systems, presented the hypotheses clearly, and summarized the study's anticipated contribution concisely. These changes contribute to a well-rounded understanding of the subject and the context in which the research was conducted.

Material and Methods Clarifications: The reviewer asked for more details on the exclusion criteria, variables used, equations introduced, and handling of missing data. The authors responded by elaborating on all these aspects, enhancing the clarity of the methodology. This transparency will likely bolster the reader's confidence in the research process.

General Compliance with Reviewer’s Suggestions: The authors seem to have addressed each of the reviewer's points comprehensively. They not only made necessary revisions but also explained their reasoning in detail.

Evaluation for Publication: Overall, the manuscript seems to have been significantly improved based on the reviewer's insightful comments, and the authors have shown great diligence in addressing the concerns raised. The study appears to be methodologically sound, and the topic is relevant and specific. Assuming the scientific quality and novelty align with the standards of the intended publication outlet, the paper seems to merit publication.

Author Response

Dear reviewer, 

We appreciate the time and effort that you dedicated to providing detailed feedback on our manuscript and are grateful for insightful comments on and valuable improvement to our paper.

We believe that our manuscript has benefited greatly from your comments.

Chiaki Toida
